# The Cap-Binding Complex CBC and the Eukaryotic Translation Factor eIF4E: Co-Conspirators in Cap-Dependent RNA Maturation and Translation

**DOI:** 10.3390/cancers13246185

**Published:** 2021-12-08

**Authors:** Jean-Clement Mars, Mehdi Ghram, Biljana Culjkovic-Kraljacic, Katherine L. B. Borden

**Affiliations:** Institute for Research in Immunology and Cancer (IRIC), Université de Montréal, Pavillion Marcelle-Coutu, Chemin Polytechnique, Montreal, QC H3T 1J4, Canada; jean-clement.mars@umontreal.ca (J.-C.M.); mehdi.ghram@umontreal.ca (M.G.); biljana.culjkovic@umontreal.ca (B.C.-K.)

**Keywords:** m^7^G cap, CBC, eIF4E, cap-chaperones, mRNA maturation, splicing, polyadenylation, nuclear export, cancer

## Abstract

**Simple Summary:**

To produce the proteins needed for the cell to survive, the information in the DNA is converted to a mobile form known as messenger RNA, which exits the cell nucleus and binds to machines that convert RNAs into proteins in a process referred to as translation. Importantly, the RNA message can be altered at any point in its journey prior to translation. These alterations constitute changes to the chemical nature of the messenger RNA and modulate the ability of mRNAs to be converted into proteins and even lead to the production of proteins with different functionalities than those encoded by their original forms. Here we provide a conceptual framework for the integration of these mRNA maturation steps with translation with a focus on the proteins that escort these mRNAs through these steps and into the translation machines. We discuss the relevance to cancer and therapeutic strategies to target these in malignancy.

**Abstract:**

The translation of RNA into protein is a dynamic process which is heavily regulated during normal cell physiology and can be dysregulated in human malignancies. Its dysregulation can impact selected groups of RNAs, modifying protein levels independently of transcription. Integral to their suitability for translation, RNAs undergo a series of maturation steps including the addition of the m^7^G cap on the 5′ end of RNAs, splicing, as well as cleavage and polyadenylation (CPA). Importantly, each of these steps can be coopted to modify the transcript signal. Factors that bind the m^7^G cap escort these RNAs through different steps of maturation and thus govern the physical nature of the final transcript product presented to the translation machinery. Here, we describe these steps and how the major m^7^G cap-binding factors in mammalian cells, the cap binding complex (CBC) and the eukaryotic translation initiation factor eIF4E, are positioned to chaperone transcripts through RNA maturation, nuclear export, and translation in a transcript-specific manner. To conceptualize a framework for the flow and integration of this genetic information, we discuss RNA maturation models and how these integrate with translation. Finally, we discuss how these processes can be coopted by cancer cells and means to target these in malignancy.

## 1. Overview

The translation of RNAs into proteins is a fundamental step in the conversion of DNA signals (in the form of coding transcripts) into their active form (proteins). Most focus has been on the impact of modulating the translation of a given RNA on the ultimate levels of the corresponding protein produced. While this is clearly important, it is equally critical to consider the processes that chemically modify the transcripts which ultimately impact the composition of the final protein product, as well as the processes that regulate the availability of these transcripts to the ribosome. In this regard, many of the major RNA metabolism steps can physically alter the composition of transcripts, which can lead to altered abilities of RNAs to access the ribosome and/or the production of different functional forms of the resulting proteins. In this way, RNA metabolism provides a means to reprogram the transcriptional signals which are the ultimate substrates of the ribosomes. For instance, splice variants and alternatively polyadenylated transcript variants can produce proteins with opposing or modified functions [1,2]. Elevated RNA nuclear export can increase the availability of transcripts to the translation machinery, bypassing the need to increase transcription [3]. Understanding how RNA metabolism segues into translation is critical for the decryption of the subsequent functional proteome which in turn governs cell physiology and, when dysregulated, can underpin human malignancy.

A fundamental feature central to the control of RNA maturation, export, and translation is the m^7^G “cap” modification on the 5′end of coding RNAs [4]. Factors that bind this moiety can control the fate of the RNA. There are two major cap-binding factors in mammals [5]. The cap-binding complex (CBC) is comprised of the cap-binding protein NCBP2 (also known as CBP20) and its adaptor NCBP1 (also known as CBP80) which together escort RNAs through the different nuclear maturation steps including splicing, CPA, and nuclear RNA export [6,7,8,9,10]. CBC also acts in the pioneer-round of translation and even in the steady-state translation of some viral and cellular RNAs [11,12,13]. The eukaryotic translation initiation factor eIF4E, which is associated with the steady-state translation of RNAs [14], can also function in capping, splicing, CPA, and nuclear RNA export for selected transcripts [15,16,17,18,19,20,21]. Indeed, both CBC and eIF4E can be found in the nucleus and cytoplasm of mammalian cells, positioning them to act in all these functions simultaneously [7,8,9,10,18,22,23,24,25]. In this way, the proteins involved in these chemical alterations to the RNAs impact their translation. This suggests that RNA maturation and translation are more closely tied than previously considered. Indeed, the classic model is that CBC escorts RNAs in the nucleus as they undergo maturation steps that will define their function, transporting them to the cytoplasm where these RNAs are handed off to eIF4E for steady-state translation [14] (Figure 1). In this model, there is a clear separation-of-function based on subcellular location; however, more recent evidence suggests this needs to be revisited given the new cellular localization and functionalities discovered for CBC and eIF4E over the past 25 years.

Here we discuss the evidence supporting the roles of these cap-chaperones in RNA maturation and translation with the goal of understanding the flow of genetic information and its impact on the cell. These observations lead us to propose that there are coexistent, parallel RNA maturation paths characterized by CBC or eIF4E cap-chaperones. Moreover, we predict that RNAs can undergo cap-chaperone switching between CBC and eIF4E. In other words, transcripts can undergo certain RNA maturation events with one cap-chaperone and then switch the cap-chaperone for subsequent steps. This model provides a means by which groups of RNAs can be differentially matured and translated in response to a wide array of stimuli and collectively dysregulated in malignancy. This review is divided into three sections: (1) a description of RNA maturation, RNA export, and translation focusing on the role of cap-chaperones; (2) models for RNA maturation based on these observations and the interplay with translation; and (3) relevance to cancer.

## 2. RNA Maturation, Nuclear Export, and Translation Focusing on the Roles of Cap-Chaperones

One of the most highly held tenets in gene expression is that during transcription, coding as well as many non-coding RNAs, undergo a series of processing steps that physically modify the chemical structure of the RNA in order to generate mature transcripts that are substrates of the ribosome [26]. There are three major steps in this process for coding RNAs: m^7^G capping, splicing, and CPA [26]. This is followed by nuclear export to the cytoplasm which can amplify the signal by increasing cytoplasmic RNA availability to the ribosome without altering transcription levels [27,28]. In some cases, translation can also be used as a source for amplification of the transcriptional signal by increasing the number of ribosomes per transcript so that more protein is made from the same amount of transcript. The specific steps and relevant biochemical reactions are described in this section, followed by the relevance of cap-chaperones CBC and eIF4E to each step. These steps often occur co-transcriptionally, enabling a transcription–RNA maturation coupling. We also discuss recent findings which indicate that many of these steps can also occur post-transcriptionally in the nucleus and, in some cases, even in the cytoplasm.

### 2.1. The Process of m^7^G Capping

The first maturation step for most transcripts is referred to as m^7^G capping [4,29,30]. The function of the m^7^G cap is multifold. It enables the engagement of specific cap-binding factors such as CBC and eIF4E and, furthermore, it protects RNAs from degradation by 5′exonucleases [31,32]. Capping is key for the efficiency of subsequent RNA maturation steps including splicing, CPA, RNA export, and translation [9]. Generally, it is considered that RNAs are capped co-transcriptionally after generation of the first ~20–30 nucleotides. In mammals, capping is a three-step process involving RNGTT (RNA guanylyltransferase and 5′ phosphatase) and RNMT (RNA guanine-7-methyltransferase) [33,34,35], two proteins essential for capping and also for cell survival [33,34,36,37,38,39,40,41]. RNGTT removes the 5′ phosphate of the 5′ triphosphate on the pre-mRNA or non-coding RNA using its 5′ phosphatase activity [42]. The resulting 5′-diphosphate-RNA serves as an RNGTT substrate for the addition of guanosine monophosphate, resulting in a 5′-5′ pyrophosphate linkage. Then, RNMT uses its methyltransferase domain and the S-adenosyl methionine (SAM) methyl donor to methylate the cap guanylate [38,43,44]. RAM, a small protein co-factor, binds RNMT and increases its methylation activity and can also recruit some RNAs to the complex [43]. At steady state, the localization of RNMT and RNGTT is mainly nuclear; however, these factors are also found in the cytoplasm supporting the notion that capping can occur in both locals [36,45,46]. It also suggests that decapped RNAs are not always fated to be degraded but rather can be re-capped [47,48,49]. Further, while capping is generally considered to be co-transcriptional, the cytoplasmic localization of these enzymes and the observation of re-capping activity in that compartment suggest that re-capping could also occur independently of transcription [45,46,50]. Thus, as with splicing and CPA [51,52,53] (see below), it appears that capping can occur both co- and post-transcriptionally depending on the RNA and cellular context. The traditional view is that shortly after capping, RNAs directly bind to the NCBP2 component of the CBC [10]. At this stage, the RNA is available to be chaperoned through the splicing, CPA, and nuclear export steps by the CBC. The role of CBC in mRNA maturation has been reviewed extensively [9,13,54].

While capping was widely considered to be a constitutive process by which all coding RNAs are 100% capped, recent findings suggest a more nuanced view should be adopted. Indeed, steady-state capping is dynamic, leading to the notion of cap homeostasis [4,47]. These findings are based on new methods developed to study capping which can be carried out on a per RNA or genome-wide basis [4,55]. These studies revealed that capping efficiency is positioned as an important control point for the protein-coding capacity of many mRNAs as well as the biochemical activity of non-coding transcripts [15,29,30,48,55,56]. Indeed, the capping of specific transcripts can be regulated during development and differentiation, and can be elevated in human malignancy [15,29,30,48,55,56]. The dynamic nature of capping is highlighted by the observation that RNAs can be de-capped and then re-capped [45,46,47,50]. The equilibrium between cap-chaperones and decapping enzymes could be modulated to transiently change the proportions of capped RNAs based on their recruitment affecting specific transcripts. It seems likely that since capping is dynamic, the cap-binding protein associated with RNAs varies depending on the cellular conditions, suggesting that transcripts could use different cap-chaperone proteins during disparate parts of their life cycle.

Recently, eIF4E was shown to play a role in steady-state capping [15]. eIF4E elevates the production of RNMT, RNGTT, and RAM through its RNA export and translation activities [15], and also physically interacts with RNMT in cells and in biophysical experiments [57]. Consistently, the overexpression of eIF4E in cell line models led to the increased capping of a subset of transcripts. Many of the eIF4E target RNAs encode proteins involved in oncogenesis, e.g., *MDM2*, *CTNNB1*, *MYC*, and *CCND1*, but not housekeeping transcripts. While eIF4E increased capping for some RNAs, it reduced the capping for others, suggesting there is a competition for limited resources (RNGTT, RAM, RNMT, SAM) so that eIF4E tips the balance to favor specific transcripts. These studies also revealed that in human cells the capping of specific RNAs at steady-state was lower than anticipated, positioning this as an important regulatory step in terms of the functionality of RNAs. eIF4E does not simply provide protection against decapping by directly binding to m^7^G caps of its target RNAs [15]. Indeed, eIF4E overexpression increases the steady-state capping of transcripts not found in eIF4E nuclear RIPs, indicating that eIF4E does not need to physically interact with these to enhance capping [15]. An RNA element, referred to as a cap sensitivity element (CapSE) conferred increased capping activity when fused to a *LacZ* reporter upon eIF4E overexpression [15] and could recruit relevant factors to the RNA. In all, eIF4E increases capping for a subset of RNAs, likely containing specific RNA elements that imbue these RNAs with sensitivity to this process by recruiting factors such as RNMT to the transcripts.

### 2.2. RNA Splicing

Splicing is the removal of introns and the joining of flanking exons in pre-messenger RNAs (pre-mRNAs) and some non-coding RNAs [58]. This process is catalyzed by the spliceosome, an intricate assembly of >150 proteins and 5 uridine-rich small nuclear UsnRNAs (U1, U2, U4, U5, and U6 snRNAs) [58]. This machine recognizes elements in the 5′ splice-site (5′SS), 3′ SS, and the branch site to catalyze the excision of targeted introns. Alternative splicing (AS) generates diversity in the proteome by producing multiple mRNAs from the same pre-mRNA [1]. About 95% of multi-exonic genes undergo AS [59]. AS events include altered selection of the 5′SS or 3′SS, skipped exons (SE), inclusion of mutually exclusive exons (MXE), or intron retention (IR) [59]. These events occur through competition between different splice sites in the pre-mRNA, where this balance is altered by cis-elements known as exon/intron splicing enhancers (ESE/ISE) or silencers (ESS/ISS) [59]. Strong ESS/ISS elements promote exon skipping whereas ESE/ISE sequences stimulate splicing. AS products can have opposing functions to their constitutive counterparts, lead to mis-localization, generate transcripts that are rapidly degraded, leading to protein loss, or other effects [1,60].

Recently, the location of splicing has been shown to be more diverse than first thought. Approximately 80% of splicing occurs co-transcriptionally and ~20% post-transcriptionally in nuclear speckles [52,61]. Features of intron location (e.g., near the 3′end, terminal introns) or intron composition (highly structured) favor post-transcriptional splicing [51,52,61,62]. Interestingly, in the same RNA, some introns can be removed co-transcriptionally with others post-transcriptionally [51,61]. In these cases, spliceosome assembly on the RNA can result from two independent assembly events (i.e., the spliceosome does not appear to travel with the RNA [61]). Additionally, cytoplasmic splicing was reported [63,64]. Indeed, ~0.5% of the transcriptome appears to undergo cytoplasmic splicing via the minor spliceosome. This complex shares many common components with the major spliceosome in the nucleus [63]. Cytoplasmic splicing has not been extensively studied and remains controversial, but the possibility of such post-transcriptional processing extends the capacity of mRNA maturation to modify transcripts and the resulting protein outputs.

The possibility that splicing occurs in different locations, co-transcriptionally and post-transcriptionally, leads to the question of whether different cap-chaperones would be used in different contexts. It is generally considered that RNAs are recruited to the spliceosome through interactions of their m^7^G cap with the CBC. CBC is most often associated with removal of the first intron in a co-transcriptional manner [65], where it triggers this event by promoting the recognition of the 5′ splice site by U1 snRNP [7]. Further, CBC binds the splicing machinery in an mRNA-independent manner [8]. For example, CBC promotes spliceosome assembly and interacts with UsnRNPs [8]. Additionally, CBC associates with U6 snRNP, which occurs concomitantly with the recruitment of the U4/U6-U5 tri-snRNP [65,66].

In addition to facilitating the constitutive splicing of first introns, CBC regulates AS through its co-factor, the positive transcription elongation factor P-TEFb, supporting a transcription-dependent regulation of AS by CBC [67]. Indeed, gene-specific recruitment of P-TEFb by the CBC [67] may explain how CBC affects AS for only a subset of transcripts. There is the proposed involvement of sequence-specific RNA-binding proteins such as SRSF1 [68], or structure-specific RNA-binding protein such as hnRNPF [69], suggesting that RNA *cis*-elements influence the CBC-dependent control of AS. Finally, CBC may have an indirect influence on AS through the regulation of snRNAs synthesis and export [70,71,72]. In all, while CBC generally associates with RNAs co-transcriptionally, it seems likely that is it not involved in all splicing events for all transcripts. This suggests that other cap-chaperones could escort pre-mRNAs through these steps. For example, in the nucleus, eIF4E can modulate the splicing of thousands of RNAs, increase the levels of components of the spliceosome, and physically interact with all the major UsnRNAs as well as protein components of the spliceosome [21]. This occurs in multiple human cell lines and in high-eIF4E acute myeloid leukemia (AML) patient specimens, suggesting that in parallel to CBC escorted RNAs, eIF4E can also chaperone a subset of RNAs to the spliceosome. It also raises the question of whether RNAs could be escorted through splicing by other cap-chaperones. However, further investigation is needed to confirm that eIF4E is associated directly with the mRNAs and not via interactions with splice factors or UsnRNA as seen for CBC. Interestingly, a recent genome-wide CRISPRi screen identified exon junction complex (EJC) and spliceosome components as lethal partners of eIF4E. The list of identified genetic eIF4E partners also includes more than twenty genes coding for proteins involved in ribonucleoprotein complex biogenesis, including ribosome biogenesis and other nuclear functions [73]. These findings clearly highlight the functional proximity of eIF4E with the nuclear pathways of RNA maturation.

### 2.3. Cleavage and Polyadenylation

Largely considered to be a co-transcriptional event, CPA is a two-step process required for maturation of the majority of coding RNAs and also for some non-coding transcripts [74]. This maturation is important for both nuclear RNA export and for ultimate translation for coding transcripts. The first step of CPA is the recognition of the polyadenylation signal (PAS) within the 3′ UTR of transcripts, which is then followed by cleavage of the polyadenylation site near to the PAS. Cleavage mainly occurs in the nucleus [74], and is followed by addition of the poly(A) tail via poly(A) polymerases which can occur in either the nucleus or cytoplasm [75,76,77]. Importantly, many transcripts have more than one PAS cleavage site, which can even occur within the coding region of the RNAs [75,76,77]. Alternative polyadenylation (APA) is defined as the cleavage of alternative PASs within the 3′UTR, introns, or coding region of transcripts; altered PAS cleavage efficiency; and/or alterations in poly(A) tail length [75,76,77]. APA can affect RNA stability, export, localization, translation, and in some cases, the functionality of the protein product by altering the coding sequence, such as by generating truncated proteins with alternative C-terminal domains [2]. The impact of APA on gene expression is highlighted by the observation that at least 70% of mammalian RNAs express APA isoforms [77].

Interestingly, even though CPA is mainly a co-transcriptional nuclear process, transcripts can undergo additional CPA either post-transcriptionally in the nucleus or in the cytoplasm [53,78,79]. Analyses of the human transcriptome has revealed that thousands of cleaved and polyadenylated transcripts can be re-cleaved at a proximal internal site in the 3′-UTR, resulting in two stable, autonomous, RNA fragments: a coding sequence with a shorter 3′-UTR (body) and an uncapped 3′-UTR sequence downstream of the cleavage point (tail) [78]. This suggests a widespread post-transcriptional APA producing stable 3′-UTR RNA tails that co-exist alongside their transcripts of origin [78]. This study also suggested that shorter transcripts are more abundant in the cytoplasm and that these events are cell-type specific [78]. Observations of the *CTN* RNA (Cat2 transcribed nuclear RNA) showcased the notion of nuclear post-transcriptional APA [53]. Here, the *CTN* RNA undergoes a first round of CPA, likely co-transcriptionally, but then the polyadenylated mRNA remains stored in the nucleus until specific signaling events trigger the cleavage of another PAS within the *CTN* RNA to produce the *CAT2* RNA, which is followed by poly(A) tail addition and is then exported to the cytoplasm and translated into protein [53]. Cytoplasmic, post-transcriptional APA in neurons provides a fascinating similar example [79]. Here, selected groups of transcripts are processed into mRNAs with long 3′UTRs in the nucleus. This selective APA seems to promote the nuclear export and the subsequent axonal transport of these mRNAs to be translated at specific sites within the neurons. In this case, longer 3′UTRs are necessary for the recruitment of RNA binding proteins (RBPs) and non-coding RNAs needed for their axonal transport and ultimate neuronal localization. Afterwards, efficient local translation requires, in most cases, shorter 3′UTRs so the cleavage of a proximal PAS is triggered at specific cytoplasmic locales [79]. One of the most interesting studies in this regard showed that hundreds of axonal transcripts with shorter 3′UTRs are produced in the cytoplasm. This cytoplasmic cleavage is mediated by a multi-protein complex centered around the endonuclease Ago2 [80].

Cap-chaperones are involved in PAS selection and can influence cleavage efficiency. CBC, for instance, specifically regulates the 3′end processing of structured transcripts from intronless genes such as replication-dependent histones (RDHs) and pre-snRNAs. CBC promotes the selection of proximal PAS through two mutually exclusive co-factors: the negative elongation factor (NELF) [81] and the arsenite resistance protein 2 (ARS2) [82,83,84]. Given its role in splicing by recruiting U1 snRNP to the 5′splice site of the first intron, CBC also inhibits the selection of some intronic PASs [83], providing a clear example of how cap-chaperones can be key contributors to the interplay between splicing and CPA. The alternative CBC component, the nuclear cap-binding protein NCBP3, also plays a role in suppressing premature CPA through its interaction with U1 snRNP [85]. Additionally, the other major cap-chaperone, eIF4E, is likely to be another regulator of CPA [19]. eIF4E physically associates with CPSF3 (the polyadenylation site cleavage enzyme) and its co-factor CPSF1 in cells and can increase the cleavage of model RNAs. It increased the cleavage efficiency of a few examined RNAs that harbored an eIF4E sensitivity element (4ESE) previously identified for eIF4E-dependent export [19]. In parallel, eIF4E drives the production of several cleavage co-factors by increasing both their RNA export and/or translation efficiency, including the cleavage enzyme CPSF3 and its co-factor CPSF1 [19]. Our unpublished results show that eIF4E overexpression or reduction alters the PAS site selection of ~1000 RNAs, suggesting a substantive role in the APA of selected transcripts.

APA contributes to disorders ranging from cancer to neurodegeneration [75,76,77]. In cancer, APA generally results in oncogenes coding mRNAs with shorter 3′UTRs, providing a mechanism to evade regulation by microRNAs or RBPs through the removal of key regulatory sequences [75,76,77]. These shorter RNAs are usually generated through the selection of PASs close to the stop codon (known as the proximal PAS). However, shorter 3′ UTRs are not always oncogenic, with effects often being context specific. APA can be dysregulated for many reasons [75]. However, the molecular mechanisms driving APA are only starting to be elucidated. Cap-chaperones, including CBC and eIF4E, are likely to be significant contributors to APA regulation as well as its dysregulation in malignancies.

### 2.4. Nuclear RNA Export

To reach the cytoplasm, mRNAs do not diffuse freely through the nuclear pore complex (NPC). Instead, mRNAs transit to the cytoplasm through the NPC escorted by NPC receptors and an array of export co-factors including cap-chaperones. Once in the cytoplasm, the mRNA cargoes are released from these factors so that they can be ultimately translated by the ribosome. NPC receptors and export factors are then reimported into the nucleus to act in future rounds of export. The addition of the cap enhanced the export of both snRNAs and mRNAs, highlighting the importance of cap-chaperones to this process [86]. Consistent with these observations, one of the protein co-factors facilitating the export of RNAs is the cap-chaperone CBC, which is associated with the m^7^G cap of mRNAs and also with the 2,2,7-trimethylguanosine cap on snRNAs during export, albeit in different complexes [87,88]. CBC-bound mRNAs associate with the bulk RNA export receptor NXF1/NXT1 [86,89] which in turn interacts with the nuclear basket of the NPC and then transits through the membrane via the central channel of NPC [90,91,92]. NXF1/NXT1 recruits mRNAs through the transcription export complex (TREX). TREX consists of UAP56, ALY/REF (ALY), CIP29, and the multi-subunit THO complex, which is comprised of THOC1/Hpr1, hTho2, THOC5, THOC6, THOC7, and Tex1 [89]. TREX is generally considered to recruit transcripts during splicing [93], and it interacts through ALY to 5′ and 3′ of transcripts [94,95]. In CBC, while NCBP2 is bound to the cap [96], NCBP1 interacts with different partners, including ALY, which in turn recruits TREX at the 5′ends of spliced mRNAs, leading to the export of the complex [97]. However, the inhibition of NCBP2 suggests that UsnRNA export needs CBC, while mRNA export does not [70], opening the possibility that other factors could play a cap-chaperone role here. The accessory CBC protein NCBP3, found to bind the m^7^G cap, albeit with less affinity than NCBP2 [6], was also identified as an interactor of TREX, suggesting a role in mRNA export [6]. Additionally, NCBP3 is associated with the exon-junction complex (EJC) and the stimulated export of polyadenylated RNAs through cooperation with TREX [98]. However, NCBP3 may not be a cap-binding protein in vivo, but rather, a protein associated with CBC, TREX, and the EJC to export spliced mRNAs [98,99,100]. Further investigation is required to dissect these possibilities.

An alternative nuclear export path for selected mRNAs employs CRM1/XPO1 to transit through the NPC rather than NXF1/NXT1 [27,28]. This pathway could utilize CBC and/or eIF4E. Indeed, eIF4E increases the export of many RNAs through the CRM1/XPO1 system. One well-defined means of selection of RNAs for this pathway is the presence of the 4ESE RNA element in the 3′UTR of transcripts which is comprised of a paired loop structure [17,18,20]. This element is recognized by LRPPRC (leucine-rich pentatricopeptide repeat C-terminus protein). The export complex formed with 4ESE-RNA is composed of eIF4E, LRPPRC, and the export receptor CRM1/XPO1 [20,101]. LRPPRC acts as an assembly platform where it interacts directly with the dorsal surface of eIF4E (bound to the m^7^G cap on the target RNA) and with CRM1/XPO1 [20,101]. The independence of eIF4E-dependent export from the bulk pathways was demonstrated by its sensitivity to leptomycin B treatment [18,102] which directly inhibits CRM1 [102] and its insensitivity to the genetic inhibition of NXF1/TAP1 [18]. Additionally, eIF4E remodels the NPC to promote 4ESE mRNA export [16]. Indeed, eIF4E overexpression leads to alterations to the NPC by reducing levels of RanBP2, the major cytoplasmic filament protein, and altering the localization of other NPC factors [16]. Moreover, eIF4E drives the expression of RanBP1, likely to compensate for the reduction in RanBP2, thereby allowing efficient RNA cargo release in the cytoplasm [16]. Intriguingly, eIF4E may also impact the bulk RNA export pathway through its ability to increase levels of Gle1 and DDX19 [16], which are required for the release of the RNA cargo into the cytoplasm in the NXF1/NXT1 pathway [16]. In this way, eIF4E is positioned to indirectly modulate the export of RNAs going through the CBC–NXF1/NXT1 pathway, highlighting an example of crosstalk between these cap-chaperone pathways. CBC uses the CRM1 pathway to export snRNAs [71], and thus it may also be able to export mRNAs similarly; this will require further studies to ascertain. The eIF4E pathway shares common elements but is distinct from, and parallel to, the CBC-directed RNA export pathway which usually employs NXF1/NXT1 [101].

### 2.5. Translation

Translation can be regulated at many steps and has been recently reviewed [103,104,105]; here, we focus on the initiation step. eIF4E-dependent translation initiation is considered as the “reference” pathway in that regard (see recent reviews [14,104,106]). In this pathway, once mRNAs are in the cytoplasm, the multi-protein translation initiation complex eIF4F assembles on the m^7^G cap of the RNA [107]. The eIF4F complex consists of eIF4E, eIF4A (an RNA helicase), and the scaffold protein eIF4G, which binds to both the 43S pre-initiation complex (PIC) and eIF4F, including eIF4E bound to the m^7^G-capped mRNA [107,108]. After assembly, the 43S PIC scans the 5′UTR until it reaches a start codon, and then recruits the 60S ribosome subunit to form the 80S ribosome, which can then start the elongation step [108]. In terms of translation, eIF4E overexpression often leads to enhanced translation efficiency, i.e., an increase of the number of ribosomes per transcript. Increases in levels of eIF4E increase the translation efficiency of specific RNAs in the cell while not altering others, such as *GAPDH* or *ACTN* [109]. In addition to its direct involvement in translation initiation, eIF4E could also impact translation via its ability to drive the expression of DDX19 and Gle1 [16], two export factors that also play independent roles in translation initiation and the termination of selected mRNAs [110,111].

Interestingly, CBC is also implicated in translation via its cap-binding activity [14]. The best described role for CBC here is in the pioneer round of translation which occurs as transcripts exit from the nuclear pore and is usually considered as a quality control mechanism to assess the readiness of RNAs for steady-state translation. The pioneer round of translation can occur using either cap-chaperone, CBC or eIF4E [95,112]. CBC-mediated translation is often associated with nonsense-mediated decay [113] or with a protein surveillance mechanism called the aggresome–autophagy pathway [114]. CBC also plays roles in steady-state translation during stress, including prolonged hypoxia, serum deprivation, or ionizing radiation [115,116,117,118,119]. For example, during hyperosmotic stress in yeast, CBC actively engages polysomes where it mediates translation of around 600 transcripts, about 10% of the transcriptome [120]. These conditions inhibit eIF4E-mediated translation [120]. During HIV infection, CBC plays a role in the translation of unspliced HIV transcripts under conditions where eIF4E-dependent translation is inhibited [11]. CBC also plays a role in the translation of specific types of RNAs, such as histone transcripts [81,121,122].

CBC translation complexes are distinct form those employed by eIF4E. For example, CBC–RNA complexes recruit the CBC-dependent translation initiation factor (CTIF), which acts as an assembly platform protein similar to eIF4G. CBC–CTIF associates with eIF3 through eIF3g to recruit the small ribosomal subunit and to access the start codon [123]. In this situation, eIF4A3 is employed as a helicase component to unwind the RNA structure while scanning [122]. Recent work suggested that CBC-dependent translation could be restricted to perinuclear localization by the association of CTIF-DDX19, suggesting that the nuclear export factor is a co-regulator of CBC-dependent translation and/or that DDX19 also plays a role in the pioneer round of translation and/or during stress.

eIF3d plays a well-established role in ribosome recruitment during translation initiation [124]. However, recently it was shown that eIF3d is a cap-binding protein and is able to drive cap-dependent, eIF4E-independent translation [125,126]. eIF3d-capped RNAs associate with an assembly factor DAP5 rather than eIF4G or CTIF. eIF3d–DAP5 targets the translation of ~20–30% of the translatome [127]. Other factors are also involved in specific forms of cap-dependent translation, such as PARN and LARP1 [14,128]. As more cap-binding proteins continue to be identified, the spectrum of cap-dependent translation pathways will undoubtedly be enlarged. Indeed, these studies suggest that eIF4E does not monopolize the translation pathway for all RNAs or for all conditions.

## 3. Models for RNA Maturation Based on These Observations and Their Interplay with Translation

As described above, the combinatorial outcomes of RNA maturation define the functional proteome. Here, we describe classic notions for how these events are coordinated and discuss the possibility of parallel RNA maturation networks governed by cap-chaperones.

### 3.1. Classical Linear RNA Maturation Model

In this model, the CBC binds RNAs during the early stages of transcription, shortly after they have been capped [10,129] (Figure 1). The CBC then chaperones the RNAs through co-transcriptional splicing [7]. This occurs contemporaneously with, or just prior to, the selection of PAS cleavage sites followed by polyadenylation. In this model, the physical changes to the RNA are now complete (capping, intron removal, and PAS site cleavage), except for variation to poly(A) tail length [130]. Next, RNAs continue to be chaperoned by CBC to the nuclear pore complex where they associate with the bulk mRNA export receptors, NXF1/NXT1, and exit into the cytoplasm where they may undergo a pioneer round of translation to check mRNA quality using either CBC or eIF4E. These RNAs are then handed off to eIF4E for steady-state translation, which leads to protein generation. Eventually, RNAs are retired through decay. CBC is considered the sole nuclear cap-chaperone in this model.

### 3.2. Non-Linear RNA Maturation Model

This model is built upon the classical linear model but incorporates recent observations that capping, splicing, and CPA can sometimes occur post-transcriptionally as well as co-transcriptionally (Figure 2). Indeed, these events do not have to occur in the nucleus. For example, re-capping, CPA, and perhaps splicing can take place in the cytoplasm for specific transcripts (as described above). Furthermore, the same transcript can undergo both co- and post-transcriptional processing [53,79]. For example, intron-containing RNAs can be polyadenylated and then undergo splicing [51]. This indicates that RNA maturation is not always a process that is simply finished once RNAs leave the transcription sites, as predicted from the classical linear model. Further, it shows that the physical changes to the RNA can occur out-of-order i.e., not only in the order of capping-splicing-CPA-export-translation. We refer to this out-of-order processing as non-linear. This model incorporates temporal and geographical aspects of CBC-based RNA maturation that are not present in the classical, linear view of RNA maturation.

### 3.3. Parallel, Non-Linear, Interleaved RNA Maturation Model

There are several observations that support the possibility that chaperoning capped RNAs through maturation steps is not limited to CBC but includes eIF4E, and by inference, other cap-binding proteins. These observations include (1) there are substantial amounts of both eIF4E and CBC found in the nucleus where most maturation occurs [9,22,23,24,131]; (2) both eIF4E and CBC bind to RNAs in the nucleus [9,16,17,18,101,132,133]; (3) both eIF4E and CBC physically associate with splicing [9,21], CPA [9,19], export [9,18,20,101], and translation machinery [14]; and (4) eIF4E and CBC have an impact on multiple steps of the maturation for specific subsets of RNAs [15,16,18,19,20,21,101]. Thus, we propose that multiple, parallel RNA maturation paths coexist, each employing different cap-chaperones (Figure 3). In this new model, CBC does not have a monopoly on flux through these nuclear RNA maturation steps, nor does eIF4E monopolize the cap-chaperone activity in the cytoplasm. RNAs can transit through the maturation steps in parallel, directed by either CBC or eIF4E cap-chaperones. Whether CBC or eIF4E both act in co-transcriptional maturation, and/or post-transcriptional maturation, remains an open question that needs to be addressed experimentally. In this way, eIF4E could provide a parallel path for selected RNAs to undergo maturation. Further, as in the non-linear model above, this model can incorporate aspects of both co- and post-transcriptional RNA maturation, i.e., it can account for out-of-order processing.

It is likely that RNAs are not restricted to one cap-chaperone path. In this case, we propose that RNAs can undergo cap-chaperone switching during their maturation and translation in a process we refer to as interleaving (Figure 3). In this model, RNAs directed to CBC-mediated processing could generate better substrates for eIF4E in subsequent steps, or vice versa. In this way, RNAs may not be restricted to CBC- or eIF4E-dependent maturation paths. It is likely that there will be substantial heterogeneity in any given RNA population, whereby conditions would cause a population shift between predominant cap-chaperones rather than an “all-or-nothing” system. In this model, the predominant cap-chaperone can vary for the same RNA population at different steps of maturation, e.g., splicing vs. APA. In support of this, *CCND1* and *MYC* transcripts are found to immunoprecipitate with both NCBP1 and eIF4E in the nucleus [101]. By contrast, *GAPDH* is restricted to associations with NCBP1 with no binding of this RNA with eIF4E detected in the nucleus [101]. Further, eIF4E overexpression drives more *CCND1* and *MYC* transcripts to exit via eIF4E-dependent export rather than bulk export mediated by CBC, while *GAPDH* is not an eIF4E-dependent RNA export target [101]. These observations support a model that some RNA species can undergo cap-chaperone switching while others may be restricted to one cap-chaperone type. Ultimately, this transformed information flow will change the proteome functionality by impacting on the availability of the transcript to the translation machinery, translation efficiency, C-terminal definition of proteins, and protein structure based on differences derived from the splice isoform. A clear prediction from this model is that CBC and eIF4E should interact to hand off RNAs. Our lab and others had previously found it difficult to observe physical interactions between eIF4E and NCBP1 [95,101,134,135]. The recent availability of NCBP2 antibodies and more sensitive Western blot reagents prompted us to revisit this question. Indeed, these advances allowed us to now observe immunoprecipitations between NCBP2 and eIF4E (Mars et al., unpublished observations). In all, cap-chaperone switching can provide a molecular basis for RNAs to be exchanged back-and-forth between eIF4E- and CBC-parallel paths at different RNA maturation steps.

Recent studies provide one possible example for how cap-chaperone switching might occur [119]. In this case, the studies were investigating the mechanisms underpinning the transition from CBC-mediated mRNA nuclear export to eIF4E-mediated translation initiation [119]. Here, the double-stranded RNA-binding protein Staufen1 interacts with the capped mRNA and facilitates the interaction between the CBC–importin-α/β complex, which leads to the dissociation of CBC from the cap and allows the recruitment of eIF4E to the RNA [119]. Interestingly, this switching process was inhibited in response to ionizing radiation to maintain CBC–mRNA association. Inhibiting the transition from CBC to eIF4E obviates eIF4E-dependent translation and therefore induces apoptosis under these conditions [119]. The regulation of cap-chaperone switching could be positioned to impact on the adaptation to stress triggers [115,116,117,118,119].

It is important to consider what underlies the preference of one cap-chaperone over another for a given transcript. Presumably, this depends on the availability of cap-chaperones and related co-factors as well as the physical nature of the RNA itself. In this latter case, the splice isoform, CPA isoform, and/or similar modifications will likely impact the recruitment or exclusion of co-factors that would then influence the subsequent processing steps available to that RNA (Figure 3). This specificity is likely informed as in the RNA regulon models by a complement of USER (untranslated sequence element of regulation) codes [136,137,138]. Since the m^7^G cap is a USER code for both CBC and eIF4E, other elements within the RNA are needed to confer further selectivity. In this model, USER codes in RNAs will engage certain steps such as splicing which can in turn modulate the USER code content to favor specific outcomes in terms of the functional protein (Figure 3).

There is evidence for cap-chaperone switching and the influence of USER codes in this process [16,17]. For example, examination of the physical associations *LacZ-4ESE* and *LacZ* RNAs with CBC and eIF4E in the nucleus revealed that both RNAs bound to CBC (but not eIF4E) in the nuclear matrix fraction; however, in the nucleoplasmic fraction, *LacZ-4ESE* only bound to eIF4E (and not CBC) while *LacZ* only bound to CBC (and not eIF4E) [101]. Thus, cap-chaperone switching occurred for *LacZ-4ESE* but not *LacZ* RNAs. Consistently, *LacZ-4ESE* was exclusively bound to eIF4E-dependent RNA export factors and was exported solely by that pathway [101]. By contrast, *LacZ* did not bind to eIF4E or its associated export co-factors in the nucleus and was exported via the CBC-dependent bulk pathway [101]. In the cytoplasm, eIF4E binds both RNAs while CBC does not, characteristic of a subsequent cap-chaperone switching step shortly after arrival in the cytoplasm [101]. In all, the 4ESE USER code was sufficient to select the eIF4E cap-chaperone path in the nucleus leading to eIF4E-dependent export [101]. In this way, the 4ESE confers a cap-chaperone preference which is also observed for some forms of eIF4E-dependent APA [19]. While the USER codes can impart selectivity for cap-chaperones, conversely the maturation steps could control whether those USER codes are present, which then influences the processing steps that are still available to that RNA (Figure 3).

Other examples of cap-chaperone switching have been observed as for the Staufen1 example above. Indeed, RNAs passed from the pioneer round of translation by CBC to steady-state translation by eIF4E in a step referred to as the “hand off” is a well characterized example of cap-chaperone switching [113,117]. Also, in support of cap-chaperone switching, nuclear eIF4E binds ~3500 RNAs but does not act in the RNA export for all of these. Thus, those bound to eIF4E at some point, must switch to CBC (or another cap-chaperone) to be exported. Indeed, it would be of interest to establish the relative fraction of RNAs bound to eIF4E and CBC to further develop this model. While eIF4E has been reported to bind to many transcripts in the nucleus, a direct assessment of the number of transcripts bound to CBC in parallel (from the same nuclear lysates) has not been conducted to our knowledge but would provide important information. While it may, at first glance, seem easy to determine the fractions of RNAs that are present in eIF4E or CBC RIPs in such an experiment, it is difficult to accurately quantitate the relative fractions bound, particularly as the relative avidity of the eIF4E and CBC antibodies would undoubtedly confound this issue. In all, experiments such as this will provide important insights into this process.

It is also important to consider that the status of the cap itself will be key for assigning which cap-chaperone(s) is favored. In this review, we focus on m^7^G caps. However, a small percentage of coding RNAs have FAD, NAD, UDP-glucuronic acid, UDP-glucosamine, m^3^G, or even potential protein caps such as the potyviral VPg protein [139,140]. While eIF4E is known to bind the viral VPg protein [139], the cap-chaperones associated with these other cap-types are not yet known. This suggests that for these more exotic caps, there will be specialized cap-chaperones which will engage other parallel paths for their maturation.

Notably, eIF4E has an additional impact on RNA maturation that has not been reported for CBC. eIF4E impacts the production of components of the capping, splicing, CPA, and export machinery [15,16,19,21]. This is driven by its ability to increase the RNA export and/or translation of many of these components. For example, eIF4E enhances the production of bulk RNA export/translation factors (e.g., *Gle1*, *DDX19*) through increased RNA export of their corresponding transcripts as well as by reducing levels of the NPC component RanBP2 indirectly [16]. eIF4E similarly stimulates the production of the capping (*RNMT*, *RAM*, and *RNGTT*), splicing, and CPA (*CPSF3*, *CPSF1*, and others) machineries [15,19,21]. This provides a basis for wide-scale reprogramming of the RNA machinery to impact “bulk processing” as well as the eIF4E maturation path. In contrast, CBC overexpression or knockdown is not reported to impact the production of the RNA maturation machinery. However, further experimentation to directly test this possibility needs to be carried out. In all, the flow of genetic information can be reprogrammed simultaneously at multiple levels by eIF4E through its cap-chaperone activity as well as its ability to modulate levels of multiple maturation factors simultaneously. Thus, eIF4E does not only co-opt information flow for eIF4E-bound transcripts but can impact other RNAs, that it is not bound to, through impacts on the production of this machinery. In all, the multipronged impacts of both CBC an eIF4E are positioned to influence the functional proteome through their direct roles in RNA maturation, nuclear export, and selective translation.

## 4. Dysregulation of Cap-Chaperones in Cancer

There is substantial evidence that RNA maturation, export, and translation can be dysregulated in many human malignancies. In cell lines, eIF4E overexpression promotes foci formation, growth in soft agar, invasion, migration, tumor formation, and apoptotic rescue from a variety of stimuli [141,142,143,144]. In xenograft mouse models, elevated eIF4E correlates with increased tumor numbers, invasion, and metastases [145,146,147,148], while pharmacological inhibition or reduction of eIF4E levels suppress tumor growth. In transgenic models of eIF4E overexpression, mice develop a variety of cancers [149,150]; while partial loss of eIF4E does not substantially impact transgenic mice, the complete deletion of eIF4E is lethal [144]. eIF4E-mediated transformation was thought to rely only on the increased translation of oncogenic mRNAs [151]. However, eIF4E’s nuclear functions are also critical for its oncogenic activities as shown by several studies. For example, an eIF4E mutant which completely impairs its translation activity but remains fully active in nuclear export (eIF4E-W73A) transforms cells as readily as wild-type eIF4E [16]. Conversely, the mutation of S53A in eIF4E impairs both RNA export and transformation but does not impede translation [23,24]. Furthermore, the nuclear localization of eIF4E is closely tied to its transformation properties. eIF4E is imported into the nucleus through a direct interaction with importin 8 via its cap-binding site [24]. Importin 8 overexpression led to increased foci formation which was lost in the case of the genetic reduction of eIF4E even though importin 8 levels remained elevated and, in turn, eIF4E lost its oncogenic activity upon the reduction of importin 8 [24]. Indeed, the addition of a nuclear localization signal (NLS) increased eIF4E’s nuclear levels and oncogenicity [24]. Further support of the relevance of the nuclear roles of eIF4E to its oncogenic potential arises from observations that eIF4E’s RNA export and oncogenic activity are inhibited by the nuclear protein PML through its direct interaction with eIF4E and its ability to impair eIF4E’s cap-binding activity in the nucleus [23,152]. Similarly, the proline rich homeodomain protein PRH (also known as Hex), but not PRH without an NLS, inhibits eIF4E-related mRNA export and oncogenic activity through direct nuclear interaction [133]. For many RNAs encoding oncogenes and related factors such as cyclin D1, GPI, hexokinase, and HAS3, eIF4E elevates their production via increased nuclear export, but not via increased translation [143,153], consistent with the relevance of nuclear eIF4E to its oncogenic activity [16,17,18,19,24,143,154,155]. Consistently, the ability of eIF4E to modify the nuclear pore is also central to its oncogenic activity [16,24]. Indeed, while eIF4E elevates NPC components such as RanBP1 and stimulates export factors such as Gle1 and DDX19, it reduces levels of the major component of the cytoplasmic fibrils of the NPC, RanBP2. The overexpression of RanBP2 inhibits both eIF4E-mediated mRNA nuclear export and eIF4E-induced cell transformation. The opposing roles of eIF4E and RanBP2 is highlighted by the observation that eIF4E overexpression reduces RanBP2 protein levels [16]. While the nuclear localization and nuclear export functions of eIF4E are drivers of its oncogenic potential, the roles of APA and AS are not yet known but are exciting areas of future study.

eIF4E levels are increased in many cancers, where its elevation generally correlates with poor prognosis [132,141,142,145,148,154,155,156,157,158]. Some cancers characterized by dysregulated eIF4E include acute myeloid leukemia (AML), multiple myeloma, diffuse large B-cell lymphoma, breast cancer, prostate cancer, head and neck cancer, and others [132,142,145,154,155,156,157,159,160,161,162,163,164,165,166]. Several of eIF4E’s activities are found to be elevated in primary patient specimens, including capping, splicing, RNA export, and translation [15,132,154,155,157,159,160]. The genetic or pharmacological inhibition of eIF4E was shown to reduce oncogenicity and has been correlated with both its nuclear and cytoplasmic functions [154,155,156,157,159,160,161,162,163,164,165,166,167,168]. For example, 4EGI-1 inhibits the formation of the eIF4E–eIF4G complex in breast cancer stem cells [168] and induces apoptosis in malignant pleural mesothelioma [169]; presumably, 4EGI-1 also inhibits the nuclear functions of eIF4E, but this remains to be tested. The direct pharmacological inhibitor of eIF4E, ribavirin [167,170,171], competes for cap-binding and thereby impairs both its RNA export and translation functions [154,158,167]. Ribavirin impairs the binding of eIF4E to importin 8 and prevents eIF4E’s nuclear entry in cell culture and in patients treated with ribavirin [24,155]. In high-eIF4E AML patients prior to treatment, eIF4E is almost entirely nuclear [132,155]. During clinical response, ribavirin blocks eIF4E’s association with importin 8, leading to impaired nuclear entry and reduced eIF4E-dependent mRNA export, which correlates with clinical responses [24,155,160]. eIF4E is mainly cytoplasmic and at relapse eIF4E is once again highly nuclear, supporting that its oncogenic potential derives at least in part from its nuclear localization [155,160]. During disease progression in patients, ribavirin becomes chemically modified and no longer binds eIF4E, allowing eIF4E to associate with importin 8 and to re-enter the nucleus leading to enhanced eIF4E-dependent nuclear RNA export [24,155,160,172]. Here the assembly of nuclear export complexes appears slower than the export and/or release of cargoes, leading to the nuclear enrichment observed for eIF4E at steady-state in untreated patients or during disease progression [24]. To date, ribavirin is the only direct eIF4E inhibitor to be used in patients. Ribavirin treatment leads to objective responses including remissions in two completed and one ongoing clinical trial in AML as well as objective responses in head and neck cancer and castration-resistant prostate cancer clinical trials [155,160,162,173]. There are >15 ongoing trials using ribavirin to target eIF4E in cancer (see ClinicalTrials.gov (accessed on 25 November 2021)). Clearly, inhibition of eIF4E activity is likely to be potent because it impacts nuclear and cytoplasmic activities and also the capacity of eIF4E to produce relevant RNA maturation and export machinery.

eIF4E has also been targeted with antisense oligonucleotides (ASOs) in animal models and patients. Unfortunately, in patients, there were no responses better than short-term stable disease despite the promising results of ASOs in prostate mouse models [174]. This failure appears to be due to the low efficiency of ASO-mediated reduction of eIF4E in humans [174], whereas this was very potent in mice [156]. Increased levels and phosphorylation of eIF4E are associated with aggressiveness in patients [141]. Our lab was first to show that the phosphorylation of eIF4E is linked to its oncogenic potential as well as its RNA export activity [175]. There are ongoing studies using inhibitors of eIF4E phosphorylation (MNK inhibitors) (ClinicalTrials.gov NCT02040558 (accessed on 25 November 2021)), the results of which are not yet known. eIF4E phosphorylation by MNK impacts its nuclear RNA export function [175,176], but its role in translation remains controversial. Moreover, MNK phosphorylates other proteins as well as eIF4E [14,177,178,179], and thus MNK inhibitors are not specific to eIF4E or translation, but more broadly impact RNA metabolism, which could underpin their prospective potency. The eIF4E partner protein and RNA export target, CPSF3, has been shown to be a potential target candidate as well as a prognostic biomarker for a multitude of cancers, e.g., AML and Ewing’s sarcoma [180,181,182]. Simultaneous inhibition of these two factors would constitute a very interesting approach to investigate the impact on countering oncogenesis, especially in the case of myeloid malignancies, amongst others. In summary, the cap-chaperone eIF4E has been largely studied in the context of oncogenesis.

By contrast, CBC is relatively understudied in terms of its role in malignancies. The depletion of NCBP2 or NCBP1 is not lethal in mammalian cells [6]. However, a recent study pointed out a critical role in oncogenic processes such as cell proliferation, invasion, and migration [183]. NCBP1 is elevated in lung adenocarcinoma and a mechanism associating NCPB1 to NCBP3 increased the expression of CUL4B, leading to oncogenic phenotypes in this context [183]. The protein atlas also identified NCBP1 (https://www.proteinatlas.org/ENSG00000136937-NCBP1/pathology (access on 25 November 2021)) as a favorable prognostic marker for prostate cancer and unfavorable for pancreatic cancer, while NCBP2 (https://www.proteinatlas.org/ENSG00000114503-NCBP2/pathology (access on 25 November 2021)) has been identified as an unfavorable prognostic marker for liver, pancreatic, and prostate cancer. Further investigations are needed to more deeply investigate the involvement of the different CBC components in oncogenesis and whether they constitute biomarkers and/or targets.

## 5. Open Questions and Conclusions

Above, we discussed the information flow that generates the transcripts that will ultimately underpin the functional proteome. There remain several open questions to further probe and build on this model. For instance, does eIF4E or other cap-binding proteins aside from CBC interact with RNAs co-transcriptionally and act in maturation at that site? Does eIF4E act in RNA maturation in the cytoplasm beyond translation? Does CBC act in post-transcriptional CPA or splicing in the nucleus or cytoplasm? What are the USER code complements that provide enrichment for specific cap-chaperones? To date, the 4ESE, in combination with the m^7^G cap, is known for favoring eIF4E-dependent nuclear RNA export and CPA; however, further decoding is necessary to properly dissect the USER codes that will underpin cap-chaperone choice and the subsequent decision tree of information flow. This information flow can have substantive impacts on cell physiology, even leading to structural and functional changes to the cell by altering the functionality of the proteome.

Our model of non-linear, parallel, interleaved RNA maturation provides a molecular basis to subvert the flow of genetic information and thus the functional proteome (Figure 3). In principle, this model is equally applicable to non-coding RNAs, and their resulting functional transcriptome. The outcome of each maturation step will impact the possibilities for the subsequent steps. In other words, CBC and eIF4E can re-write the RNA message depending on the relative outcomes of each maturation step, thereby partially decoupling the original messages derived from the DNA from the ultimate translated product. Furthermore, it suggests that drugging cap-chaperones could lead to multiple simultaneous impacts on RNA maturation, export, and translation, potentially increasing their potency. Thus, understanding the complexity and organization of the information flow derived from parallel RNA maturation paths to translation will have important therapeutic implications. Further experimental investigation is ultimately required to test this model and will undoubtedly lead to a better understanding of how these processes are integrated to generate the functional proteome.

## Figures and Tables

**Figure 1 cancers-13-06185-f001:**
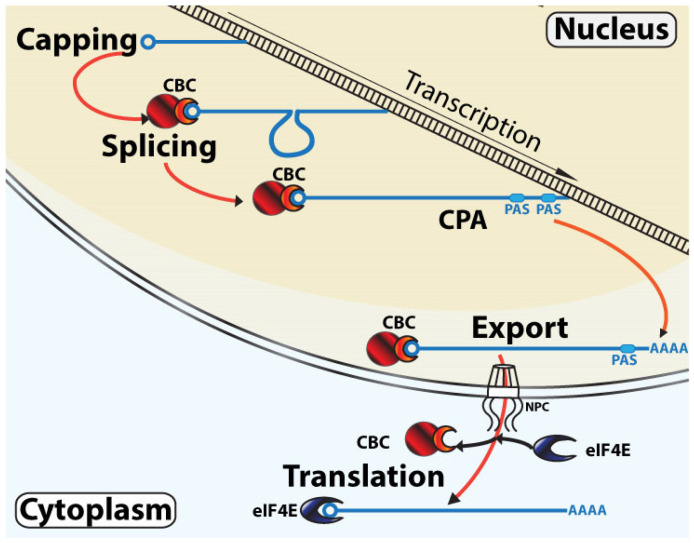
Classical linear model of mRNA maturation. During transcription, the m^7^G cap is added to the RNA as it is transcribed. This chemical modification leads to CBC recruitment. While transcription elongation continues, CBC facilitates the recruitment of the spliceosome to the nascent RNA. At the end of transcription, CBC assists in the recruitment of the cleavage and polyadenylation (CPA) machinery at the PAS site. The CBC chaperones the mRNA to the NPC where the complex transits through the nuclear membrane and into the cytoplasm where eIF4E replaces the CBC as the predominant cap-chaperone. The transcript can now undergo steady-state translation. The m^7^G cap is depicted as a circle, CBC is in red with NCBP1 as the larger circle and NCBP2 as the semi-circle, and eIF4E is in blue. For simplicity, not all factors involved in these processes are shown.

**Figure 2 cancers-13-06185-f002:**
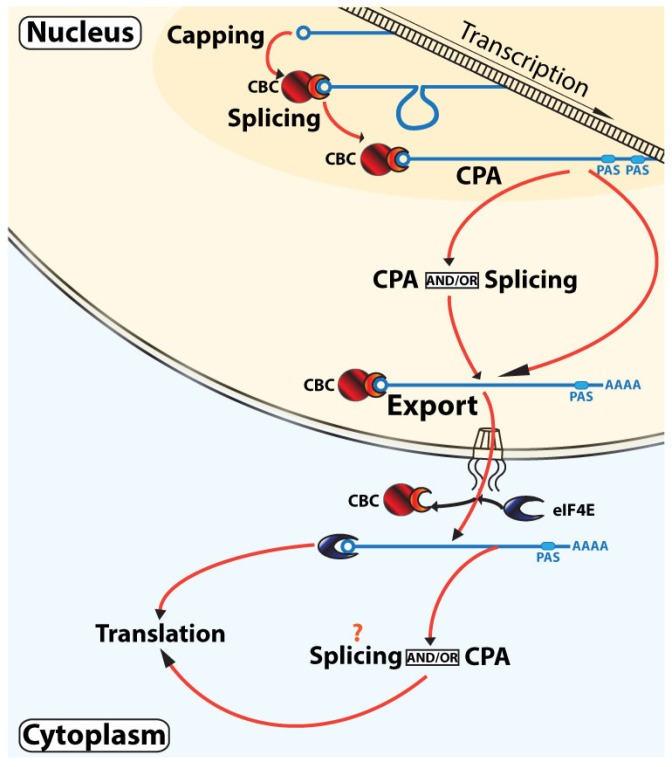
Nonlinear model of RNA maturation. This model incorporates recent studies that demonstrate that splicing and CPA can take place both co-transcriptionally (darker orange highlighted region) and/or post-transcriptionally in the nucleus (lighter orange highlighted region) or in the cytoplasm (light blue). Once exported to the cytoplasm, the transcriptome can still be modified by capping, CPA, and perhaps splicing. As described in the text, CBC-mediated translation occurs but is not depicted here for simplicity. The m^7^G cap is depicted as a circle, CBC is in red with NCBP1 as the larger circle and NCBP2 as the semi-circle, and eIF4E is in blue. For simplicity, not all factors involved in these processes are shown.

**Figure 3 cancers-13-06185-f003:**
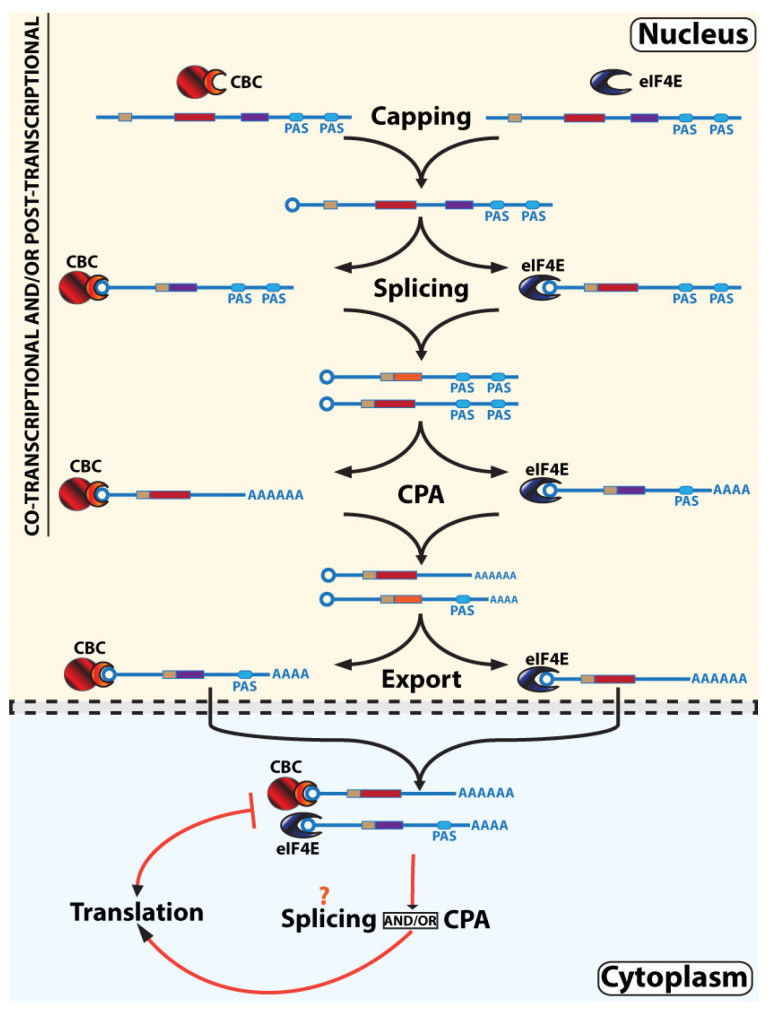
Parallel, non-linear, interleaved RNA maturation model. We propose a model whereby multiple, parallel RNA maturation paths coexist, each employing different cap-chaperones. RNAs can interleave between eIF4E- or CBC-dependent pathways by a process we refer to as cap-chaperone switching. For simplicity, we focus here on demonstrating these aspects of the model but note that as for Figure 2, these processes can occur co- and/or post-transcriptionally and can be nuclear or cytoplasmic. The evidence for interactions of both CBC and eIF4E with these steps is described in the text. The different PAS sites (blue ellipses) and splicing events (exons in different coloured rectangles) are depicted to highlight our hypothesis that differing RNA compositions lead to different complements of USER codes, and this in turn will influence whether CBC or eIF4E is favored and under certain circumstances drive cap-chaperone switching as depicted in the figure. The m^7^G cap is depicted as a circle, CBC is in red with NCBP1 as the larger circle and NCBP2 as the semi-circle, and eIF4E is in blue. For simplicity, not all factors involved in these processes are shown.

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
