# Peer review of "The Cap-Binding Complex CBC and the Eukaryotic Translation Factor eIF4E: Co-Conspirators in Cap-Dependent RNA Maturation and Translation"

_cancers, 2021, doi:10.3390/cancers13246185_

Round 1

Reviewer 1 Report

I thought this was a very nice review that summarises the role of cap binding proteins in RNA metabolism including roles beyond translation.  This will certainly be of interest to readers of this journal.  It starts by setting out the background, before moving into alternative and more provocative models before exploring the contributions to cancer biology.  I thought the manuscript was well structured and well-written in clear English. I would support publication in its current form.

This is a review of the role of cap binding proteins in RNA metabolism.  A common view is that these proteins function exclusively in translation.  This review explores their important role in all aspects of RNA biogenesis showing their contribution to splicing, polyadenylation and nuclear export.

This is not primary research it is a review of already published material.  However the extended role of cap binding proteins is not widely appreciated.

This review explores different models for cap binding protein involvement in RNA metabolism.  It does this in a balanced way by presenting three contrasting possible models that they term classical, non-linear and parallel.  Individual data papers tend to present only their own favourite model.

I think the review manuscript is publishable as it is.

The alternative models are presented and discussed based on the published literature.

Figures are helpful in supporting the explanation of differing models

Author Response

We thank the reviewer for their supportive comments.

Reviewer 2 Report

This review by Mars and colleagues provides a comprehensive overview about the various RNA processing steps and how these are regulated by the the cap binding chaperones CBC and eIF4E. The review is well written and timely. Compared to other comparable reviews on this topic the authors particularly highlight recent data on the contributions of eIF4E, which is most well-known for its canonical function in translation initiation, on the RNA maturation steps.

This reviewer found the article of great interest but felt that it was sometimes unclear to what extend certain models discussed in this review where partially speculative vs founded on direct evidence. Maybe the authors could clarify some of the examples highlighted below and also address some the following minor comments:

  • While it is clear that eIF4E exists in the nucleus and can bind capped transcribed there it is somewhat unclear to what extend transcripts are bound by eIF4E in the nucleus. Is there any quantitation on the fractions of transcripts bound by CBC vs eIF4E? It would be helpful if you comment on such evidence (or the absence of it) in the review. The same also applies to the cap-chaperone switching model.
  • Line 196: cytoplasmic splicing is a somewhat controversial 
    topic. I would suggest to cite the original paper (PMID: 21078998) and refer to it with caution.
  • Line 211-218: It is unclear whether there is direct evidence that eIF4E chaperones RNA to the spliceosome or whether eIF4E  regulates splicing rates indirectly through regulation of expression of splicing factors or binding to spliceosome components that are already bound to transcripts. Maybe the authors can clarify/comment on this?
  • Line 275: Please provide a definition of "4ESE" and "USER" elements for non-expert readers.
  • Line 301: Could you provide short description how the snRNA cap differs from the mRNA cap?
  • The authors refer to RNA editing in the lay abstract which is somewhat confusing as capping, splicing and polyadenylation are not traditionally referred to as RNA editing but rather RNA processing or maturation steps (as opposed to RNA methylation etc.)
  • Line 450-468 (related to first comment): Is there any data how frequent parallel, non-linear, interleaved RNA maturation may occur and what proportion of transcripts may be subject to this form of maturation vs the other two models? If not, it would be interesting to hear your thoughts/hypothesis?
  • Line 484 (related to first comment): the evidence cited in this review show that capped mRNAs can be bound by either CBC or eIF4E but to this reviewer it is unclear whether there is direct evidence of actual cap-chaperone switching in the course of RNA maturation. Can you please clarify this and perhaps state that the interleaving model requires further experimental investigation?
  • Line 583-584: This is a rather general statement. Can you briefly elucidate what experiments this is conclusion is based on?
  • Line 591-592: Similar as above, can you provide more specific information here? Which nuclear and cytoplasmic functions have been reported to correlate with reduce oncogenicity in response to pharmacological inhibition of eIF4E?
  • Line 626-627: which studies? Only one cited in subsequent sentence.
  • Line 632: Can you speculate why eIF4E seems to have
    a greater impact in oncogenesis compared to CBC although their functions seem to greatly overlap based on the literature referred to in this review article? Could this be due eIF4E's major role in translation initiation as compared to CBC?
  • Potential further discussion point: It wold be interesting to comment on the recent eIF4E genome-wide genetic interaction study for by the Ruggero lab and search for genetic interactions with the nuclear RNA processing machinery or CBC (PMID: 34192540). Such genetic interactions may further strengthen the link of eIF4E with nuclear RNA processing events and overlapping chaperone functions with CBC.

Minor formatting issues:

  • Line 266: remove gap between ref and "."
  • Line 322: remove gap before reference.
  • Line 330: format references.
  • Line 364: remove "." before reference.
  • Line 370: grammar - add "s" to word "play"
  • Line 374: remove gaps before the two references.
  • Line 374: Some words seem to be missing from second sentence.
  • Line 585: format references.
  • Lines 609-615: format references.
  • Line 619: format reference.

Author Response

Reviewer 2

This review by Mars and colleagues provides a comprehensive overview about the various RNA processing steps and how these are regulated by the the cap binding chaperones CBC and eIF4E. The review is well written and timely. Compared to other comparable reviews on this topic the authors particularly highlight recent data on the contributions of eIF4E, which is most well-known for its canonical function in translation initiation, on the RNA maturation steps.

This reviewer found the article of great interest but felt that it was sometimes unclear to what extend certain models discussed in this review where partially speculative vs founded on direct evidence. Maybe the authors could clarify some of the examples highlighted below and address some the following minor comments:

  • While eIF4E exists in the nucleus and can bind capped transcribed there it is somewhat unclear to what extend transcripts are bound by eIF4E in the nucleus. Is there any quantitation on the fractions of transcripts bound by CBC vs eIF4E? It would be helpful if you comment on such evidence (or the absence of it) in the review. The same also applies to the cap-chaperone switching model.
    • This is a very important and interesting question. While eIF4E has been reported to bind up to 3500 transcripts in the nucleus, a direct assessment of the number of transcripts bound to CBC in parallel (from the same nuclear lysates) has not been done to our knowledge and would be an important question to address. While it may, at first glance, seem easy to quantify the different amounts of RNAs that are present in eIF4E or CBC RIPs, it is difficult to accurately quantitate the relative fractions bound, particularly as the relative avidity of the eIF4E and CBC antibodies would undoubtedly confound this issue. We have added a comment on line 228-230.
    • The cap-switching model is inferred from the observations but is indeed a model. We thought we had made the model aspect clear but try to do so now throughout. However, there is evidence to support this model. There are examples which we discussed in the review where RNAs undergo cap-chaperone switching of CBC or eIF4E based on the presence of specific RNA elements. Also, the principle of cap-chaperone switching is apparent when RNAs go from the pioneer round of translation (using CBC) to steady state translation (using eIF4E or other related factors) in the cytoplasm. We clarified the text appropriately. See lines 524-534.
  • Line 194-196: cytoplasmic splicing is a somewhat controversial 
    topic. I would suggest citing the original paper (PMID: 21078998) and refer to it with caution.
    • We agree, the reference has been added and we discuss it in the text as controversial. We now added a question mark in the figures about cytoplasmic splicing (Figures 2 and 3)  

  • Line 217-222: It is unclear whether there is direct evidence that eIF4E chaperones RNA to the spliceosome or whether eIF4E regulates splicing rates indirectly through regulation of expression of splicing factors or binding to spliceosome components that are already bound to transcripts. Maybe the authors can clarify/comment on this?
  • We observe that eIF4E binds to splicing factors and all the UsnRNAs as well as elevated their production. Further, we observe that alterations in eIF4E levels drives widescale changes to the splicing of 1000s of RNAs. Given the immunoprecipitation with eIF4E of SF components, including markers of the active spliceosome, eIF4E is positioned to act as a cap chaperone through splicing. However, we agree with the reviewer that we cannot distinguish between the possibilities that eIF4E is acting via interactions with SFs rather than directly via capped RNAs. Indeed, CBC interacts with the spliceosome directly (does not require the mRNA). We mention this limitation in the text.

  • Line 190 and 541: Please provide a definition of "4ESE" and "USER" elements for non-expert readers.
    • We provide the definition of the 4ESE at the first occurrence which is eIF4E sensitivity element.
    • We now provide the definition (Untranslated Sequence for Regulation) as per Jack Keene’s Regulon model.

  • Line 313-316: Could you provide short description how the snRNA cap differs from the mRNA cap?
    • We now state: “Consistent with these observations, one of the protein co-factors facilitating the export of RNAs is the cap-chaperone CBC which is associated with the m7G cap of mRNAs and also with the 2,2,7-trimethylguanosine cap on snRNAs during export albeit in a different complex.”

  • The authors refer to RNA editing in the lay abstract which is somewhat confusing as capping, splicing and polyadenylation are not traditionally referred to as RNA editing but rather RNA processing or maturation steps (as opposed to RNA methylation etc.)
    • As suggested, we replaced edit by maturation to clarify the lay abstract

  • Line 546-562 (related to first comment): Is there any data how frequent parallel, non-linear, interleaved RNA maturation may occur and what proportion of transcripts may be subject to this form of maturation vs the other two models? If not, it would be interesting to hear your thoughts/hypothesis?

We observe that cap-chaperones are switched based on studies with the 4ESE element in model RNAs. In this case, in the nucleoplasm LacZ-4ESE RNAs exclusively bound to eIF4E while LacZ RNAs (without the 4ESE) are only CBC bound. However, in the nuclear matrix fraction, both RNAs bound to CBC and not to eIF4E. Thus, the LacZ-4ESE RNAs underwent cap-chaperone switching (although this term had not yet been coined).  This published work is discussed in the text.

Furthermore, there is a well-established cap-chaperone switching (referred to as the “hand off”) between CBC and eIF4E for translation in the cytoplasm.

Given that nuclear eIF4E can bind ~3500 RNAs coupled with the observations that eIF4E directly acts in nuclear export but not for all these same RNAs strongly supports the notion that while these are bound to eIF4E at some point, they become CBC bound to be exported. We now add this to the text.

We certainly agree that this is very much in the model stage. However, given that eIF4E binds so many RNAs, but does not impact on all aspects of their processing, it seems that is may be reasonably frequent. It will certainly take further experimentation to quantify the extent of this.

  • Line 524 (related to first comment): the evidence cited in this review show that capped mRNAs can be bound by either CBC or eIF4E but to this reviewer it is unclear whether there is direct evidence of actual cap-chaperone switching during RNA maturation. Can you please clarify this and perhaps state that the interleaving model requires further experimental investigation?
    • Please see the above section for the data that is known and a more complete description in the text (524-534). We completely agree that this model requires further investigation and hope to spark interest in this area through the review. We thought that was clear but tried to stress the need for further investigation.

  • Line 623-651: This is a rather general statement. Can you briefly elucidate what experiments this is conclusion is based on?
    • As suggested, we have discussed published data that support nuclear eIF4E contributes to its oncogenic functions as follows: “However, eIF4E’s nuclear functions are also critical for its oncogenic activities as shown by several studies. For example, an eIF4E mutant which completely impairs its translation activity but remains fully active in nuclear export (eIF4E-W73A), trans-forms cells as readily as wildtype eIF4E[16]. Conversely, mutation of S53A in eIF4E impairs both RNA export and transformation but does not impede translation[23,24]. Furthermore, the nuclear localization of eIF4E is closely tied to its transformation properties. eIF4E is imported into the nucleus through a direct interaction with Importin 8 via its cap-binding site[24]. Importin 8 overexpression led to increased foci formation which was lost in the case of genetic reduction of eIF4E even though Importin 8 levels remained elevated[24]. Indeed, addition of a nuclear localization signal (NLS) increased eIF4E’s nuclear levels and oncogenicity[24]. Further support of the relevance of the nuclear roles of eIF4E to its oncogenic potential arises from observa-tions that eIF4E’s RNA export and oncogenic activity are inhibited by the nuclear protein PML through its direct interaction with eIF4E and its ability to impair eIF4E’s cap-binding activity in the nucleus[23,152]. Similarly, the proline rich homeodomain protein PRH (also known as Hex) but not PRH without an NLS inhibits eIF4E-related mRNA transport and cell transformation through direct nuclear interaction[133]. For many RNAs encoding oncogenes and related factors as cyclin D1, GPI, Hexokinase and HAS3, eIF4E elevates their production via increased nuclear export, but not increased translation[143,153], consistent with the relevance of nuclear eIF4E to its oncogenic activity[16-19,24,143,154,155]. Consistently, the ability of eIF4E to modify the nuclear pore is also central to its oncogenic activity[16,24]. Indeed, it was clearly shown that, while eIF4E elevates NPC components like RanBP1, and stimulates export factors Gle1 and DDX19, the major component of the NPC cytoplasmic fibrils. Overexpression of RanBP2 inhibits eIF4E-mediated mRNA nuclear export and eIF4E-induced cell trans-formation. The opposing roles of eIF4E and RanBP2 is highlighted by the observation that eIF4E overexpression reduces the levels of RANBP2 protein levels[16]. While the nuclear localization and nuclear export functions of eIF4E are drivers of its oncogenic potential, the role of APA and AS are not yet known but are exciting areas of future study.”
    • (Lines 652-674) We also added this with regard to pharmacological inhibition: “eIF4E levels are increased in many cancers where its elevation generally correlates with poor prognosis[132,141,142,145,148,154-158]. Some cancers characterized by dysregulated eIF4E include acute myeloid leukemia (AML), multiple myeloma, diffuse large B-cell lymphoma, breast cancer, prostate cancer, head and neck cancer and others[132,142,145,154-157,159-166]. Several of eIF4E’s activities are found to be elevated in primary patient specimens including capping, splicing, RNA export and translation[15,132,154,155,157,159,160]. Genetic or pharmacological inhibition of eIF4E was shown to reduce oncogenicity and has been correlated with both its nuclear and cytoplasmic functions[154-157,159-168]. For example, 4EGI-1 inhibit the formation of eIF4E-eIF4G complex in breast cancer stem cells[168] or induce apoptosis in malignant pleural mesothelioma[169], presumably 4EGI-1 also inhibits the nuclear functions of eIF4E but this remains to be tested. The direct pharmacological inhibitor of eIF4E, ribavirin[167,170,171] competes for cap-binding and thereby impairs both its RNA export and translation functions[154,158,167]. Ribavirin impairs binding of eIF4E to Importin 8 and prevents eIF4E’s nuclear entry in cell culture and in patients treated with ribavirin[24,155]. In high-eIF4E AML patients, eIF4E is almost entirely nuclear. During clinical response eIF4E is cytoplasmic and at relapse eIF4E is once again highly nuclear supporting its oncogenic potential derives at least in part from its nuclear localization[155,160]. During disease progression in patients, ribavirin has become chemically modified and no longer binds eIF4E allowing eIF4E to associate with Importin 8 and re-enter the nucleus. In parallel, enhanced eIF4E-dependent nuclear RNA export is observed, where assembly of nuclear export complexes is slower than the export and/or release of cargoes leading to the nuclear enrichment observed for eIF4E at steady-state in untreated patients or during disease progression[24].”

  • Line 652-674: Similar as above, can you provide more specific information here? Which nuclear and cytoplasmic functions have been reported to correlate with reduce oncogenicity in response to pharmacological inhibition of eIF4E?
    • See response to above query.

  • Line 709-717: which studies? Only one cited in subsequent sentence.
    • The phrase has been re-formulated: “However, a recent study pointed out a critical role in oncogenic processes like cell proliferation, invasion and migration. » 
    • We also add the following sentence to support the need for CBC studies in cancer : “The protein atlas also identified NCBP1 (https://www.proteinatlas.org/ENSG00000136937-NCBP1/pathology) as favorable prognostic marker for prostate cancer and unfavorable for pancreatic cancer while NCBP2 (https://www.proteinatlas.org/ENSG00000114503-NCBP2/pathology) has been identified as unfavorable prognostic marker for liver, pancreatic and prostate cancer.”

  • Line 708: Can you speculate why eIF4E seems to have a greater impact in oncogenesis compared to CBC although their functions seem to greatly overlap based on the literature referred to in this review article? Could this be due eIF4E's major role in translation initiation as compared to CBC?
    • In essence, eIF4E’s dual role in nuclear and cytoplasmic compartments could undoubtedly play a role. It also seems that for nuclear export, splicing and capping functions that eIF4E impacts many oncogenic factors, presumably due to the content of USER codes within these RNAs. It is also clear that less studies have investigated the role of CBC in cancer, treating it more as a housekeeping protein. It may be that further investigation will reveal a more substantial role for CBC in future. However, it is most striking to us that eIF4E can modulate the levels of the capping, splicing, CPA and export machinery while CBC does not appear to have this capacity. This is probably the biggest difference between them. We mention this in the cancer section briefly (708-719).

  • Potential further discussion point: Its wold be interesting to comment on the recent eIF4E genome-wide genetic interaction study for by the Ruggero lab and search for genetic interactions with the nuclear RNA processing machinery or CBC (PMID: 34192540). Such genetic interactions may further strengthen the link of eIF4E with nuclear RNA processing events and overlapping chaperone functions with CBC.
    • In the splicing section we now state: Interestingly, a recent genome- wide CRISPRi screen identified Exon Junction Complex (EJC) and spliceosome components as lethal partners of eIF4E. The list of identified genetic eIF4E partners also includes more than twenty genes coding for proteins involved in ribonucleoprotein complex biogenesis, including ribosome biogenesis and other nuclear functions. These findings clearly highlight the functional proximity of eIF4E with the nuclear pathways of RNA maturation and processing to a similar extent of eIF4E-related translation regulation.

 Minor formating issues :

  • Line 266: remove gap between ref and "."
    • corrected
  • Line 322: remove gap before reference.
    • corrected
  • Line 330 : format references.
    • corrected
  • Line 364: remove "." before reference.
    • corrected
  • Line 370: grammar - add "s" to word "play"
    • corrected
  • Line 374: remove gaps before the two references.
    • corrected
  • Line 374: Some words seem to be missing from second sentence.
    • Corrected: “These conditions inhibit eIF4E-mediated” has been replaced by “These conditions inhibit eIF4E-mediated translation”
  • Line 585: format references.
    • Corrected
  • Lines 609-615: format references.
    • Corrected
  • Line 619: format reference.
    • Corrected

We have also modified the text for grammar and clarity throughout as necessary.

We hope this has addressed the reviewer’s important concerns.

Sincerely,
